# Popularity Prediction of Online Contents via Cascade Graph and Temporal Information

**Yingdan Shang, Bin Zhou, Ye Wang \*, Aiping Li, Kai Chen, Yichen Song and Changjian Lin**

College of Computer, National University of Defense Technology, Changsha 410073, China;
altsuzy@hotmail.com (Y.S.); binzhou@nudt.edu.cn (B.Z.); liaiping@nudt.edu.cn (A.L.);
chenkai_@nudt.edu.cn (K.C.); songyichen14@nudt.edu.cn (Y.S.); lcj007@nudt.edu.cn (C.L.)
\* Correspondence: ye.wang@nudt.edu.cn

**Abstract:** Predicting the popularity of online content is an important task for content recommendation, social influence prediction and so on. Recent deep learning models generally utilize graph neural networks to model the complex relationship between information cascade graph and future popularity, and have shown better prediction results compared with traditional methods. However, existing models adopt simple graph pooling strategies, e.g., summation or average, which prone to generate inefficient cascade graph representation and lead to unsatisfactory prediction results. Meanwhile, they often overlook the temporal information in the diffusion process which has been proved to be a salient predictor for popularity prediction. To focus attention on the important users and exclude noises caused by other less relevant users when generating cascade graph representation, we learn the importance coefficient of users and adopt sample mechanism in graph pooling process. In order to capture the temporal features in the diffusion process, we incorporate the inter-infection duration time information into our model by using LSTM neural network. The results show that temporal information rather than cascade graph information is a better predictor for popularity. The experimental results on real datasets show that our model significantly improves the prediction accuracy compared with other state-of-the-art methods.

**Keywords:** popularity prediction; information cascade; deep learning; social network analysis





## 1. Introduction

Online social platforms such as Facebook, Twitter, Sina Weibo and Tiktok, promote and widen the spread of online contents by attracting an increasing number of active users. A cascade of information diffusion is formed when an online content (e.g., a tweet, a microblog, or a video) is retweeted and propagated among users. The popularity of an online content is usually measured by the number of users participating in its spreading process, i.e., the number of users in the cascade graph. Predicting the future popularity of online content is greatly valuable and practically meaningful in many fields. For example, predicting the occurrence of large cascades as early as possible can create great commercial value in the field of commodity promotion and prevent the wide spread of fake news in the field of social security. Therefore, the popularity prediction task has been a hot research topic and widely used in many real-world applications, e.g., campaign strategy [1], social media recommendation [2] and viral marketing [3].

However, predicting the future popularity of online content is a challenging task since the information diffusion process is affected by many complicated factors and fraught with uncertainty. Researchers [4,5] have demonstrated that popularity prediction can be made with higher accuracy with the observation of early spread of online content, thus existing popularity prediction methods generally try to seek out the predictive features in the early state of information cascades and model the relationship between these predictive features and future popularity.

Inspired by recent successful application of deep learning in many fields, some researchers try to predict popularity through end-to-end deep learning methods. Compared with traditional approaches, deep learning based approaches for popularity prediction task [6,7] can automatically learn the complex factors affecting information diffusion without time consuming feature engineering and strong assumptions of underlying diffusion mechanism, and have achieved significant improvement of prediction accuracy. As a powerful approach for representation learning on graph data, graph neural networks [8] have been successfully applied to the field of social network analysis [9–11]. Early users involved in the cascade graph and early popularity have been proven to be highly related to future popularity [4,5,12]. Therefore, the efficient modeling of cascade graph characteristics is crucial to popularity prediction task. Recent approaches generally utilize graph neural networks to obtain low dimensional representation of information cascade graph, and model the non-linear relationship between the learned low dimensional representation and future popularity. However, existing graph neural network based popularity prediction methods confront some critical shortcomings:

Firstly, existing simple graph pooling strategies, e.g., average or summation operations treat all users in the information cascade graph equally [7,13,14], but in reality, some users in the information diffusion process are more important than others, while some users are less relevant and have little effect on further cascade propagation. Therefore, existing graph pooling strategies normally generate inefficient cascade graph representation which have limited ability to express the information spread process, and lead to poor accuracy for popularity prediction.

Secondly, existing graph neural network based methods [7,13–16] generally overlook the temporal information such as the time interval between two infections or reshares in the propagation process. However, temporal information [5,12,17] is a key predictive feature for popularity prediction task. Intuitively, a smaller mean time interval between two reshare event usually means that the information item itself is more popular in nature.

To address the aforementioned questions, we propose a novel framework integrating the cascade graph information and temporal information to predict the future popularity of online contents. We generate an effective cascade graph representation by integrating graph neural network with a novel graph pooling strategy. We focus on the top k important users and exclude the noise caused by other irrelevant users during graph pooling process. Meanwhile, we incorporate the inter-infection duration time information into our model by using Long Short Term Memory (LSTM) network [18]. By comparison of model variants, we come to a conclusion that temporal information is a better predictor for popularity prediction than cascade graph information. Our main contributions are listed as follows:

- We integrate the graph neural network framework—GraphSage [19] with a novel graph pooling methods—top-K pooling [20,21], and obtained a more effective cascade graph representation which improve the general capabilities of our model to capture the underlying diffusion characteristic.
- We incorporate the inter-infection duration time information into our model by using Long Short Term Memory (LSTM) network, and make up for the deficiencies of existing graph neural network based approaches.
- The experimental results on two publicly available real-world datasets show that our proposed method can significantly improve the cascade prediction accuracy compared to several state-of-the-art competitive baselines.

## 2. Background and Related Works

Predicting the popularity of online content is a hot research topic in recent years. Lots of researchers have studied the changing popularity of online content, such as short texts [22], photos [23], videos [24,25], etc. Traditional popularity prediction methods mainly include feature-based approaches and generative approaches. Feature-based approaches identify and extract hand-crafted features, including content features [26,27], structural features [22,28], temporal features [12,29], etc., and use machine learning algorithms to

make predictions. This type of approaches usually require heavy feature-engineering and their performance strongly depends on the effectiveness of extracted features. Generative approaches devote to model the diffusion process by probabilistic statistical generative approaches, e.g., epidemic models [30,31] and point processes [17,32–34], but the performance is normally limited by its strong assumptions of underlying diffusion mechanisms.

With the rapid development of deep learning and its successful applications in many fields, many researchers have applied deep learning technology to social network analysis [35–37], such as rumor detection, spam detection, popularity prediction, etc. Cao [38] uses coupled graph neural networks to model the global social network which refers to friendship graph and the sequence of early infected users, while assuming that information propagation usually occurs along the social network structure between users. However, in many scenarios, due to the privacy protection policies of social network platforms, it is difficult for us to obtain the global social network. Meanwhile, affected by the recommendation mechanism of the platform, users may also participate in the dissemination of information that is not posted by their friends. Therefore, some recent deep learning based popularity prediction works [6,7,14] resort to the observed early social responses, i.e., early cascade graph characteristics and early temporal information in the diffusion process, in order to improve the prediction accuracy. Information cascade graph is formed in the process of information diffusion, in which nodes represent users participating in the propagation and edges represent the propagation relations among users. Cascade graph reflects the diffusion status under the influence of many complicated factors, and it is important to model the cascade graph characteristics in popularity prediction task. Below we introduce the recent deep learning based approaches which utilize cascade graph and temporal information to make popularity prediction respectively.

## 2.1. Cascade Graph Representation

Compared with traditional methods, deep learning based approaches can learn the complicated underlying diffusion patterns in an automatic way without heavy feature engineering and strong assumption of underlying diffusion mechanisms. As a pioneer of deep learning based approaches in popularity prediction, DeepCas [39] is an end-to-end deep learning method which utilize random walk method to sample paths from cascade graph in the context of global graph structure. It also use attention mechanism to integrate the cascade path embedding into cascade graph embedding for the cascade size prediction task. DeepHawkes [6] improves the performance of popularity prediction by taking time decay effect into account when integrating the cascade path embedding into cascade embedding. Liu [40] adopts random walk processes similar with DeepCas, and generates cascade graph representation from propagation path embedding. This kind of approaches learn the cascade graph representation based on the level of propagation paths rather than the cascade graph level. Due to inefficient local structural embedding, they can not generate the effective cascade graph embedding which lead to poor prediction accuracy.

Recent studies [7,13–16] focus on learning the cascade graph level representation using graph neural networks. Recurrent cascades convolutional network—CasCN [7] propose to model the cascade graph with Graph Convolutional Network [41] and use LSTM neural network to model the dynamic changing characteristic from the sequence of cascade graph snapshots. Similarly, Cascade2vec [13] use Graph Residual block to learn the cascade graph representation and employ recurrent neural network to learn the temporal dependencies between cascade graph snapshots. Graph Attention Network (GAT) [42] is used in DMT-LIC [15] which makes micro and macro prediction simultaneously, DMT-LIC captures both the information of cascade graph and infected user sequence in the diffusion process. However, cascade graph representation in these graph neural network based methods is usually generated by a simple graph pooling strategy such as average or sum operation, which cannot distinguish user importance and results in inefficient cascade graph representation.

### 2.2. Temporal Representation

Existing deep learning based approaches often focus on learning the changing representation of cascade which grows over time but overlook the time series information such as the time interval between two infections during propagation. In DeepHawkes [6], temporal information is implicitly taken into account when integrating the cascade path embedding into cascade embedding by taking consideration of time decay effect. Researches [7,13] focus on modeling the dynamic changing characteristic from the sequence of cascade graph snapshots while neglecting the temporal information. Zhou [14] utilizes the inter-infection duration as the edge weight of cascade graph to incorporate temporal information with structural information simultaneously, but it is an artificially formulated manner with an implication that shorter infection time means closer relationship between users, but in reality, the infection-time duration does not only depend on the strength of the relationship between users but also may be caused by random factors. The methods mentioned above ignore the innate temporal pattern in the diffusion process which is a key predictive feature for popularity prediction tasks. Consequently, we exploit the explicitly temporal information in our model and demonstrate the importance of temporal information experimentally.

## 3. Materials and Methods

### 3.1. Problem Definition

We formulate the cascade prediction task as a regression problem which aims at predicting the number of users who will participate in the propagation of an online content, i.e., the size of information cascade. For an online content such as a tweet $c_i$, we denote its cascade graph at observation time $t_o$ as: $\mathcal{G}_i^{t_o} = \{\mathcal{V}_i^{t_o}, \mathcal{E}_i^{t_o}\}$. $\mathcal{V}_i^{t_o}$ is the user set who have taken part in the cascade (e.g., retweet the tweet), $\mathcal{E}_i^{t_o}$ denotes the interact relationship (e.g., retweet, comment) between users. We use $\mathcal{T}_i^{t_o} = \{t_1, ..., t_{|\mathcal{V}_i^{t_o}|}\}$ to represent the temporal information when users take part in the cascade, $|\mathcal{V}_i^{t_o}|$ represents the number of users who have taken part in the cascade at observation time $t_o$.

Given cascade graph information $\mathcal{G}_i^{t_o}$ and temporal information $\mathcal{T}_i^{t_o}$ of cascade $c_i$ at observation time $t_o$ (e.g., 1 h), our goal is to make prediction of the incremental popularity $P_i$ after a fixed time interval $\Delta t$ (e.g., 1 day), where $P_i = |\mathcal{V}_i^{t_o + \Delta t}| - |\mathcal{V}_i^{t_o}|$. Figure 1 shows a simple illustration of our problem, given early diffusion information including cascade graph and temporal information, we want to learn a regression function that maps the early diffusion information to its incremental popularity:

$$f : \{\mathcal{G}_i^{t_o}, \mathcal{T}_i^{t_o}\} \to P_i. \tag{1}$$

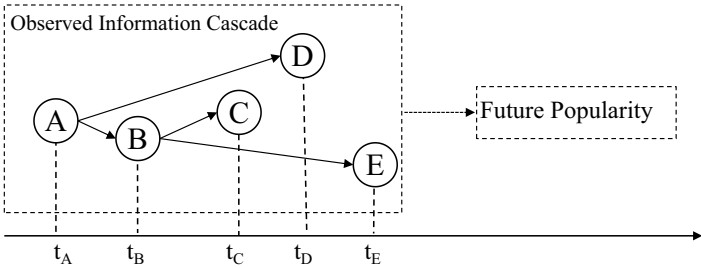

**Figure 1.** A simple illustration of problem definition.

### 3.2. Methods

In this section, we demonstrate the details of our model and explain the way we utilize the cascade graph information and temporal information to make popularity prediction. Existing studies [7,13] generally generate cascade graph representation by taking the average or summation value of all node representations, which fails to discriminate between

different users. We adopt a novel graph pooling strategy to distinguish the importance of users in the graph pooling process and improve the ability to represent the characteristic of cascade graph. Besides, we model the temporal information in the early stage of information diffusion which is often neglected by existing deep learning methods.

Figure 2 gives the overall framework of our proposed model. Our model consists of three main components: (a) Cascade graph representation: we incorporate the graph neural network GraphSage [19] with a novel graph pooling method to get the integrated cascade graph representation. In the graph pooling process, we emphasize the important users and exclude the noise brought by less relevant users by supervised learning the importance coefficient for each user. (b) Temporal representation: we use the widely adopted LSTM model and sample mechanism to generate the temporal representation from the inter-infection duration time information. (c) Predictor: as a regression problem, we concatenate the cascade graph representation and temporal representation, and feed the embedding into multi-layer perceptrons (MLPs) to make the popularity prediction.

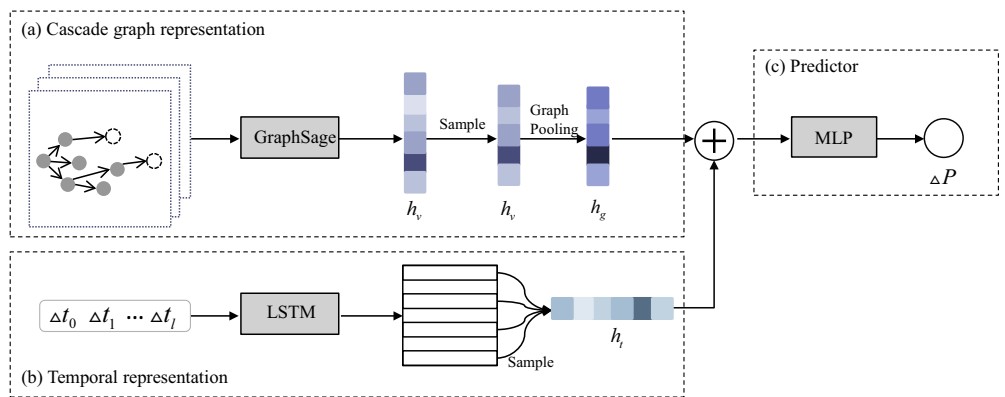

**Figure 2.** An overview of our proposed model. (**a**) Cascade graph representation: converts cascade graph information into low dimensional representation, i.e., cascade graph representation; (**b**) Temporal representation: learns temporal information based on LSTM neural network; (**c**) Predictor: maps cascade graph representation and temporal representation to popularity.

### 3.2.1. Cascade Graph Representation

Given an online information item $c_i$, and its cascade graph $\mathcal{G}_i^{t_o} = \{\mathcal{V}_i^{t_o}, \mathcal{E}_i^{t_o}\}$, we want to learn an effective cascade graph representation which reflect the actual characteristics of information diffusion. Graph neural networks are an effective deep learning framework for representation learning of graph data, we utilize the graph neural network—GraphSage as the graph convolutional layer to learn the representation of each node and a novel graph pooling method to generate cascade graph representation from node representation.

Each node $v$ in cascade graph is initially represented as a one-hot vector $\vec{q}_v \in R^N$, where $N$ is the total number of users in the dataset. Then all nodes are converted to a low-dimensional dense representation by a randomly initialized embedding matrix $E \in R^{D \times N}$:

$$\vec{h_v} = E\vec{q}_v, \tag{2}$$

$D$ is an adjustable dimension. Note the node embedding is supervised learned during the training of the model.

The node embedding learning process in GraphSage has two steps: aggregate information from neighborhood and update its own embedding. Firstly, each node $v \in \mathcal{V}_i^{t_o}$ aggregates the representation of its immediate neighborhood. We employ the max pooling aggregation strategy which can effectively capture different aspects from its neighborhood,

$$\vec{h}_{\mathcal{N}(v)}^k = max(\{\sigma(\mathbf{W}_p^k \vec{h}_u^{k-1} + \vec{b}_p^k), \forall u \in \mathcal{N}(v)\}), \tag{3}$$

$\mathcal{N}(v)$ is the neighborhood set of node v, $\vec{h}_u^{k-1}$ refers to the embedding of node $u$ at the $k-1$ graph convolutional step, $\mathbf{W}_p^k, \vec{b}_p^k$ are learnable parameters of neural networks which define how to agrregate information from neighborhood in the k-th convolutional layer, $\sigma$ refers to the sigmoid function. In the original work of GraphSage, the author uniformly samples a fixed-size set of neighbors instead of using all neighbors in order to keep the training batch size fixed. However, in our scenario, the number of node's neighbor generally obey power-law distribution. Using fixed-size set of neighbors is not a rational choice which might lead to severe information loss. Thus we use the full set of neighborhood by taking advantage of an effective GNN framework pytorch-geometric [43].

Then, we update the node embedding $\mathbf{h}_v^k$ using equation below:

$$\mathbf{h}_v^k = \sigma(\mathbf{W}^k \cdot Concat(\mathbf{h}_v^{k-1}, \mathbf{h}_{\mathcal{N}(v)}^k)), \tag{4}$$

*Concat*() means the concatenation of embeddings, $\mathbf{W}^k$ is the weight matrix used to update node embedding in the k-th convolutional layer.

Figure 3 shows us a simple example of graph pooling process. Cascade graph representation are usually generated from node representation through graph pooling process such as average or summation pooling operations. Existing graph pooling strategies for popularity prediction treat all users in the information cascade graph equally and cannot distinguish user importance. Some users are more important than others in the information diffusion process, this reality motivates us to adopt a different graph pooling strategy for the generation of cascade graph representation. Top-K pooling is one of the graph pooling strategy [20,21] which has been demonstrated to be effective on many graph classification benchmarks. The downsampling method proposed in top-K pooling within the pipeline of graph neural network is analogous to image downsampling used in CNNs, helps us emphasize the important users and excludes the noise brought by irrelevant users. In order to decide which nodes should be dropped, a learnable vector $\vec{p}$ is introduced in and an importance coefficient is calculated across the node set by the equation:

$$s_v = \frac{\vec{h}_v^k \vec{p}}{\|\vec{p}\|}. \tag{5}$$

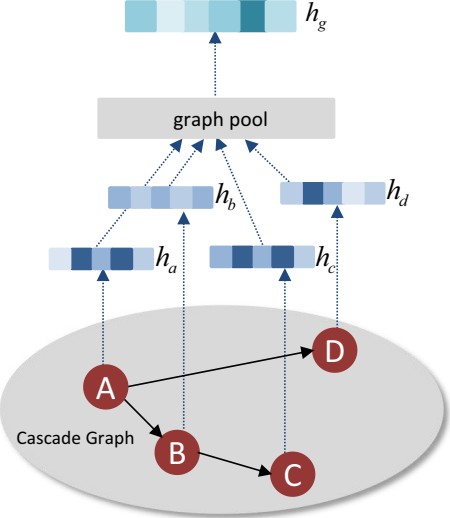

**Figure 3.** An example of graph pooling process. Cascade graph representation $h_g$ is generated from node representation by graph pooling process.

We select the top-ranked nodes from the cascade graph according to the importance coefficient $s_v$ and drop the nodes with low coefficient by a predefined downsample ratio $r$. We denote the induced subgraph as $\mathcal{G}'$. We use the learned importance coefficient $s_v$

as user weight and perform global max pooling on the induced subgraph to generate the cascade graph representation $\vec{h}_g$:

$$\vec{h}_g = max(\{tanh(s_v)\vec{h}_v^k, \forall u \in \mathcal{G}'\}). \tag{6}$$

### 3.2.2. Temporal Representation

Temporal information in the early stage of diffusion process has been proven to be a salient predictor of future popularity in traditional approaches [5,12]. However, recent graph neural network based methods normally overlook this type of information and lead to suboptimal prediction results.The time interval between two propagation events reflects the diffusion speed of information. Shorter propagation time interval usually means that the information itself is more attractive and able to trigger wider diffusion. For each online content, we extract the inter-infection duration time $\mathbf{T}_i^{t_o} = \{x_1, x_2, ...x_{|\mathcal{V}_i^{t_o}|}\}$, $x_i = t_i - t_{i-1}$ from its time series information $\mathcal{T}_i^{t_o}$. We feed the inter-infection duration time into the LSTM [18] neural network which is a variant of RNN to tackle the long-term dependency problem, and obtain a series of outputs $[\vec{o}_1, \vec{o}_2, ...\vec{o}_{max}]$ after each recurrent computation:

$$\vec{o}_t = LSTM(\vec{h}_{t-1}, \vec{x}_t), \tag{7}$$

where $\vec{x}_t$ degenerates into a scalar value, $\vec{h}_{t-1}$ is the hidden state of previous time step, $max$ is the max time step of all cascades and $\vec{o}_i \in R^d$, $d$ refers to the dimension of hidden states. We can use the final hidden states as our temporal information embedding, however, the output at different time step also contain useful information. The temporal information of first several infections are important features for popularity prediction [14], therefore, we should pay more attention on the temporal information of these early participants. Since the cascade size of online content usually follows a pow-law distribution, we sample the output sequence using the index sequence generated by $\{n * 10^m, n \in \{1, 2, ... 10\}, m = \{0, 1, 2\}\}$, and use a weighted sum to get the temporal representation from the outputs of LSTM neural network:

$$\vec{h}_t = \sum_{j=0}^{N-1} o_j \cdot \vec{\alpha}_j, \tag{8}$$

where $N$ is our sample size, $\vec{\alpha}$ is the attention vector learned automatically during training. The weighted sum mechanism can better model time decay effect during the information propagation than predefined time decay functions like exponential functions.

### 3.2.3. Predictor

The final component of our model is the predictor layer, which maps the learned predictive features in the early cascade data to its future popularity. Specifically, it takes the cascade graph embedding and temporal embedding learned in the previous step as the input, and output the future popularity of the cascade. We concatenate the cascade graph embedding $\vec{h}_g$ and temporal embedding $\vec{h}_t$ as the representation for cascade $C_i$ and feed the representation into a two-layer Multiple Layer Perception (MLP):

$$\hat{P}_i = MLP(Concat(\vec{h}_g, \vec{h}_t)). \tag{9}$$

Similar as previous work, we use the mean square log-transformed error (MSLE) as the loss function:

$$\mathcal{L}(P, \hat{P}) = \frac{1}{N} \sum_{i=0}^{N-1} (log_2 P_i - log_2 \hat{P}_i)^2, \tag{10}$$

N is the total number of cascades, $\hat{P}_i$ is the predicted increment popularity, $P_i$ is the actual increment popularity for cascade $C_i$. We take log-transformation for the cascade size following previous work to avoid the situation where the training process is dominated

by large cascades. Model parameters are trained by minimizing the loss function and optimized using the Adam algorithm given its efficiency and ability to avoid overfitting.

## 4. Experiments and Results

### 4.1. Datasets

We evaluated the effectiveness and generalizability of our model on popularity prediction task using two real-world datasets. Following many previous works [6,7,14], we adopted mean squared log-transformed error (MSLE) as our evaluation metric to measure the discrepancy between the actual popularity values and predicted values. Sina Weibo is a popular twitter-like microblog platform, the dataset provided in paper [6] is a publicly available real-world dataset, which has been widely used in many recent related works [7,13,15]. We used this dataset to predict the final retweet number of microblog. HEP-PH [7] is a paper citation dataset and was used to predict the citation count of paper.

- Sina Weibo: The dataset contained all microblogs posted on 1 June 2016, all their retweets and the corresponding retweet time within 24 h were recorded. The node in the cascade graph was the user who retweeted the microblog, and the edge between users represented their retweet relationship. Following previous works, we filtered out tweets posted in the midnight since they usually gained less attention due to less active users online. We also dropped microblogs whose retweet number was less than 10 or more than 1000 within the observation time window, because large cascades were rarely few in number and might have dominated the training process.
- HEP-PH: The dataset included paper citation relationship and paper publication time from January 1993 to April 2003. The node in the cascade graph represented the paper, and the edges referred to the corresponding citation relationship.

The detail statistics of datasets are shown in Table 1. For different observation time window T, we list out the total number of cascades, nodes, and edges in our datasets. As the observation time window T enlarged, we got more qualified cascades whose size was more than 10 and less than 1000. We also calculated the average cascade size representing average number of users participating in each cascade within the observation time.

**Table 1.** Descriptive statistics of two datasets. T refers to the observation time window.

| Datasets | Sina Weibo | | | HEP-PH | | |
|---|---|---|---|---|---|---|
| T | 1 h | 2 h | 3 h | 3 years | 5 years | 7 years |
| Number of cascades | 51,287 | 61,448 | 66,798 | 9409 | 10,629 | 10,983 |
| Number of nodes | 1,740,500 | 2,190,604 | 2,431,607 | 25,973 | 27,566 | 28,051 |
| Number of edges | 3,404,975 | 4,454,060 | 5,028,177 | 189,590 | 255,159 | 284,016 |
| Average cascade size | 66.39 | 72.49 | 75.27 | 20.15 | 24.01 | 25.86 |

As we can see from Figure 4, the distribution of cascade size followed a power-law distribution, and most information cascades' size was small. Figure 5 shows the average cascade size at different time normalized by the final average cascade size. As we can see, the microblog's popularity already reached 80% after 10 h since it was posted, so we set our observation time to be 1 h, 2 h, and 3 h when the normalized popularity was less than 60% to avoid saturation within the observation time window. We chose the observation time for HEP-PH dataset to be 3 years, 5 years and 7 years for the same reason.

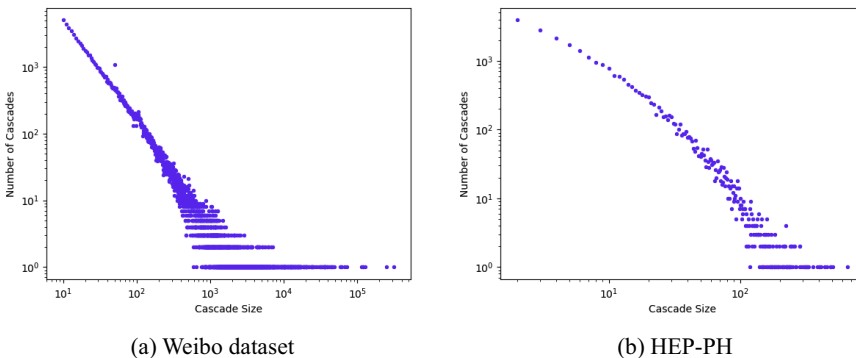

(a) Weibo dataset

(b) HEP-PH

**Figure 4.** Distribution of popularity: the *X* axis refers to the cascade size and the *Y* axis is the number of cascades corresponding to the cascade size.

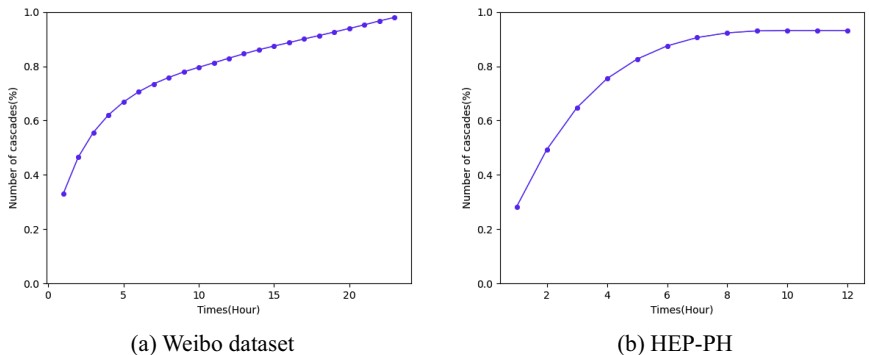

(a) Weibo dataset

(b) HEP-PH

**Figure 5.** Saturation ratio: the *X* axis refers to the time and the *Y* axis is the percentage of cascade size.

We calculated the distribution of the mean inter-infection time and cascade size for the microblog dataset. Mean inter-infection time was defined as $\frac{1}{M}\sum_{j=1}^{M}(t_j - t_{j-1})$, where M is the number of users in the cascade. Figure 6a illustrate the distribution of mean inter-infection time. As shown in Figure 6b, most large cascades (cascade size was large than 100) had a quite small mean inter-infection time—less than 4 min. Furthermore, we calculated the actual ratio of large cascades based on mean inter-infection time in Figure 6c. More than 90% large cascades in the microblog dataset were propagated with a mean inter-infection time less than 300 s. The statistical results showed that temporal information such as mean inter-infection time was highly correlated with future popularity.

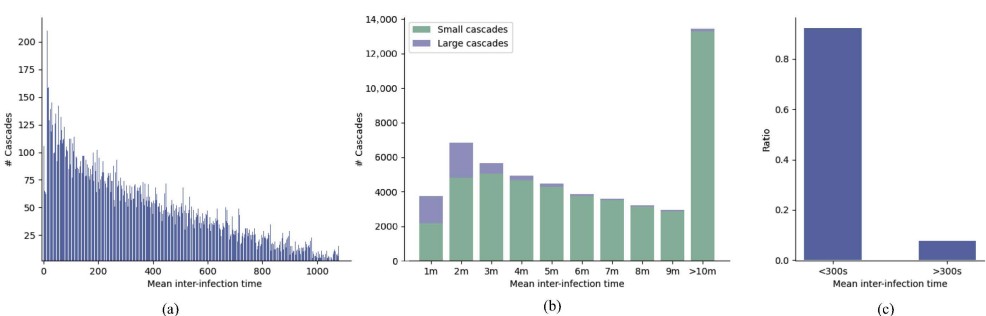

(a)

(b)

(c)

**Figure 6.** (**a**) Mean inter-infection time (seconds) and its corresponding number of cascades; (**b**) The distribution of large cascades over different mean inter-infection time (minutes); (**c**) More than 90% large cascades with a mean inter-infection time less than 300 s.

### 4.2. Baselines

We compared the performance of our method with several state-of-art deep learning based approaches including the following methods:

- DeepCas [39]: is an end-to-end deep learning method which extracts structural information of cascade graph by taking random walk in the context of global graph, and use bi-GRU neural network for the cascade size prediction task.
- DeepHawkes [6]: bridges the gap between deep learning and self-exciting point process by learning the cascade graph structural representation based on the level of propagation paths and takes time decay effect into consideration when integrating path representation into cascade representation.
- CasCN [7]: demonstrates the effectiveness in applying the graph neural network framework to generate the representation of cascade graph. It claims to exploit both the temporal and structural information by extracting cascade subgraphs from cascade graph and using LSTM neural network to model the dynamic change of cascade graphs.

The hyper-parameters of baselines kept the same as those used in model CasCN [7]. As for our model, the embedding dimensionality of nodes was 32, the batch size was set to be 16. The dimension of cascade graph embedding was 32, and temporal embedding was 16. The learning rate for our model was $5 \times 10^{-3}$. We used two graph convolutional layers in the component of cascade graph representation.

### 4.3. Variants

We provided three variants of our model to investigate the effectiveness of each component. The variants were generated by removing some part from the proposed model. We demonstrated the contribution of each component by comparing their prediction accuracy experimentally.

- VGraph (mean pool): We removed the temporal representation component from our model and only used the cascade graph representation alone. We also replaced the top-k pooling method with mean pooling method from the cascade graph representation component. The mean pooling method used the average of the embedding of all nodes in the cascade as the cascade graph embedding.
- VGraph: We removed the temporal representation component from our model and only used the cascade graph representation alone.
- VTemporal: We removed the cascade graph representation component and only used temporal representation component alone.

### 4.4. Performance Comparison

4.4.1. Model vs. Baselines

Table 2 shows the experimental results of baselines and our proposed model on two datasets. Following previous work [6,7,13], we chose mean square log-transformed error (MSLE) as the evaluation metric. In general, the performance of our model was better than the baselines on the evaluation of MSLE, surpassing the sub-optimal CasCN by around 11%. When comparing different observation time window T, we can see that as the observation time window enlarged, the MSLE error decreased, because more information was revealed to the model. The results demonstrated that our model could effectively exploit both temporal information and cascade graph information to predict the future popularity.

**Table 2.** Overall performance comparison.

| Datasets | Weibo Dataset | | | HEP-PH | | |
|---|---|---|---|---|---|---|
| **Metric** | **MSLE** | | | | | |
| **T** | **1 h** | **2 h** | **3 h** | **3 years** | **5 years** | **7 years** |
| DeepCas | 2.958 | 2.689 | 2.647 | 1.765 | 1.538 | 1.462 |
| DeepHawkes | 2.441 | 2.287 | 2.252 | 1.581 | 1.470 | 1.233 |
| CasCN | 2.242 | 2.036 | 1.910 | 1.353 | 1.164 | 0.851 |
| Proposed | 1.931 | 1.813 | 1.770 | 1.251 | 1.147 | 0.673 |

As early models which apply deep learning to cascade prediction tasks, DeepCas and DeepHawkes focused on the representation learning of the propagation path, and the ability of learning global cascade information was weak, the prediction results were not very satisfactory compared to recent graph neural network based approach CasCN.

CasCN modeled the cascade graph with a graph convolutional network and used LSTM neural network to model the dynamic changing characteristic from the sequence of cascade subgraphs, but also overlooked the importance of temporal information. Our proposed model explicitly modeled the inter-infection duration time information by using LSTM neural network, and reduced the prediction error by around 11% comparing to CasCN.

We list the performance of our model variants on the evaluation of MSLE in Table 3. In order to compare the performance of baselines and our model variants more intuitively, we also present a line chart in Figure 7. As we can see, our variants VGraph which utilized graph neural network to capture cascade graph information achieved a better prediction result compared with DeepCas and DeepHawkes, demonstrating the advantage of graph neural network based approaches.

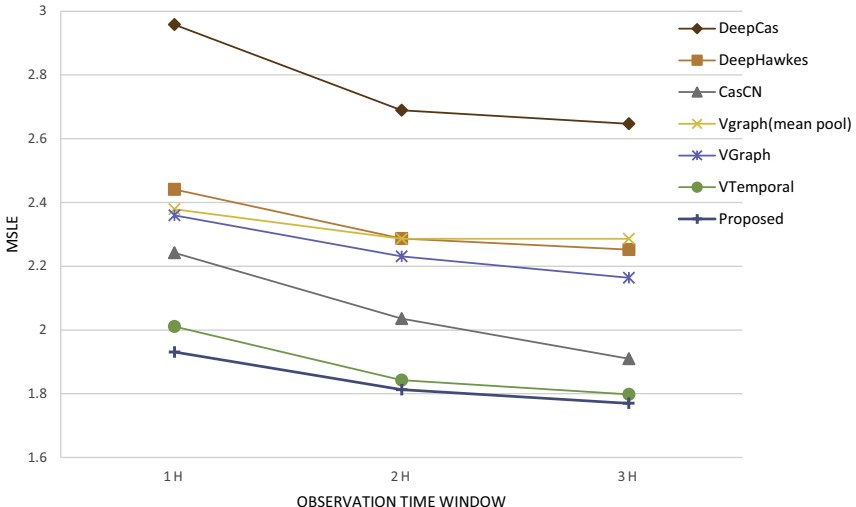

**Figure 7.** the line chart of performance comparison between baselines and model variants.

As shown in Table 3, the performance of our variant VGraph had a sub-optimal prediction result compared with CasCN, demonstrating the importance of modeling the changing characteristic of cascade graph. For model simplicity, we used the representation of cascade graph instead of dynamic cascade subgraphs and left modeling the dynamics as future work.

**Table 3.** Performance comparison of baselines and model variants.

| Datasets | Weibo Dataset | | |
|---|---|---|---|
| T | 1 h | 2 h | 3 h |
| Baseline | | | |
| DeepCas | 2.958 | 2.689 | 2.647 |
| DeepHawkes | 2.441 | 2.287 | 2.252 |
| CasCN | 2.242 | 2.036 | 1.910 |
| Variants | | | |
| VGraph (mean pool) | 2.379 | 2.286 | 2.207 |
| VGraph | 2.360 | 2.231 | 2.164 |
| VTemporal | 2.011 | 1.843 | 1.798 |
| Proposed | 1.931 | 1.813 | 1.770 |

We leveraged the temporal information in our model and made up for the deficiencies of existing methods. The model variant—VTemporal outperformed CasCN with an approximately 9% drop of MSLE error. The comparison of model variant—VTemporal and CasCN demonstrated that temporal information rather than cascade graph information was a better indicator for predicting future popularity, and confirmed the notably importance of temporal information in popularity prediction task. By jointly model the cascade graph information and temporal information, the prediction accuracy of our proposed model was significantly improved compared with state-of-art deep learning based methods.

### 4.4.2. Variants Comparison

The model variant VGraph (mean pool) adopted mean pooling strategy to generate the cascade graph representation, using the whole set of nodes in the cascade without differentiating node importance. In comparison of VGraph (mean pool) and VGraph, we demonstrated the effectiveness of top-k graph pooling strategy with a performance improvement of around 2%. We also compared the experiment results on different downsample ratio $r$ and found its best value was around 0.9 which makes intuitive sense, since noisy nodes were much less than useful nodes in reality, getting rid of too many nodes could lead to severe information loss.

The model variant VTemporal proved the significant importance of temporal information in cascade popularity prediction task. The results were consistent with previous study [44] which only utilized temporal convolutional networks to capture the temporal information for popularity prediction of messages on social medias. Further on, as Figure 7 shows, model variant VTemporal achieved a much better prediction accuracy compared with variant VGraph, and confirmed that temporal information was a better predictor of popularity than cascade graph information. The results were consistent with the past work of Shulman [5], who proved features of early adopters were weak predictors of popularity compared to temporal features.

### 4.4.3. Latent Representation

End-to-end deep learning approaches on popularity prediction are often not interpretable. In order to have a more intuitive understanding of the learned representation in our model, we used t-SNE [45] to visualize the relationship between the learned representation and cascade size in Figure 8. T-SNE was mostly used to visualize high-dimensional data and project it into low-dimensional space. Cascade graph representation and temporal representation of cascades in test dataset generated by our model were projected into points in a 2D plane. Different colors were used to distinguish cascade size, the darker color point represented the larger cascade size. The distribution of color indicated the strength of connection between the learned representation and the cascade size, i.e., if points spatially close to each other shared the same color, it meant the representation was well correlated with the cascade size. A clustering phenomenon could be seen from both cascade graph representation and temporal representation in Figure 8, indicating that both cascade graph information and temporal information were useful for cascade popularity prediction. We also notice that the clustering phenomenon of temporal information was clearer than that of cascade graph information, confirming that temporal information was more relevant with popularity and a better predictor for popularity than cascade graph information.

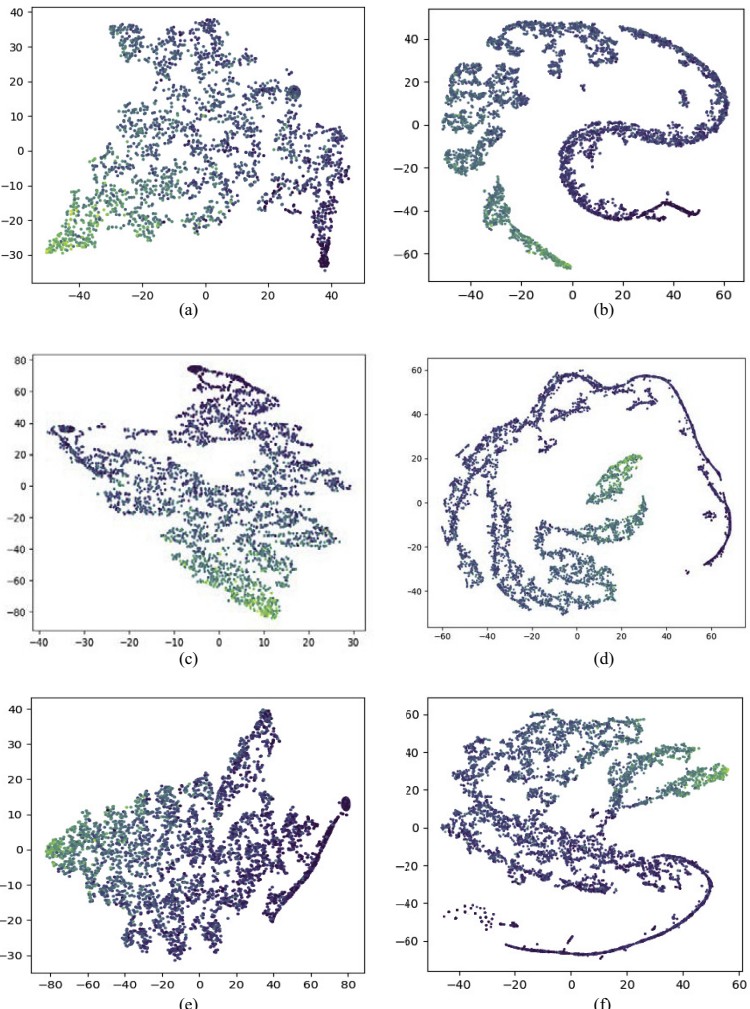

**Figure 8.** Visualization of learned representation: (**a**,**c**,**e**) show the cascade graph representations with observation time window set to be 1 h, 2 h, 3 h separately. (**b**,**d**,**f**) show the temporal representations with observation time window set to be 1 h, 2 h, 3 h separately.

## 5. Conclusions and Future Work

In this article, we propose a novel deep learning model to predict the popularity of online content by modeling the early cascade graph information and temporal information. To distinguish the importance of users and improve the ability to represent the cascade graph characteristics, we utilize the graph neural network with a novel graph pooling method. We also incorporate temporal information into our model by using LSTM neural network and make up for the deficiency of existing graph neural networks based popularity prediction methods. Experimental evaluations on two real-world datasets show that our model significantly improves the accuracy of popularity prediction comparing with other state-of-the-art methods. The experimental results demonstrate the notable importance of temporal information in popularity prediction and provide us an intuitive guidance for the subsequent popularity prediction work. In the future, it may be an interesting research point to explore the relationship between the micro-level prediction task which aims at predicting the next infected user and the popularity prediction task. Additionally, analyzing the diffusion pattern of information on different topics and predicting the popularity based on topic information may be a possible research direction.

**Author Contributions:** conceptualization, Y.S. (Yingdan Shang) and B.Z.; data curation, K.C., Y.S. (Yichen Song) and C.L.; formal analysis, Y.S. (Yingdan Shang) and Y.W.; funding acquisition, B.Z.; methodology, Y.S. (Yingdan Shang) and Y.W.; project administration, Y.W.; resources, A.L.; software,

Y.S. (Yingdan Shang); supervision, Y.W.; validation, Y.S. (Yingdan Shang) and B.Z.; writing—original draft, Y.S. (Yingdan Shang); writing—review and editing, Y.S.(Yingdan Shang) and Y.W. All authors have read and agreed to the published version of the manuscript.

**Funding:** This work was supported by the National Key R&D Program of China (No.2018YFC0831703).

**Institutional Review Board Statement:** Not applicable.

**Informed Consent Statement:** Not applicable.

**Data Availability Statement:** Weibo Dataset used in this study is included in the published articles "DeepHawkes". The corresponding supplementary information files can be found below: https://doi.org/10.1145/3132847.3132973 (6 November 2017). HEP-PH dataset is a publicly archived datasets can be found below: http://snap.stanford.edu/data/cit-HepPh.html (23 August 2003).

**Conflicts of Interest:** The authors declare no conflict of interest.

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
