# Peer review of "Popularity Prediction of Online Contents via Cascade Graph and Temporal Information"

_axioms, doi:10.3390/axioms10030159_

Round 1
Reviewer 1 Report
"Recent deep learning based approaches can effectively2model the complicated mechanism of information diffusion" they do not model the mechanism, but the non-linear (and complex) associations. If there is a paper showing how this mechanism can be inferred from the fitted tree please provide a reference.
From the deep learning techniques, Graph Convolutional Neural Networks deal with 'graphs', do you mean in the abstract the convolutions? If so you are not talking about simple DL, please be precise.
l.46 mechanisms
l.47 applications
Please provide more description of the GNN in l.54 since it is a competing method.
-Aljohani, Naif Radi, Ayman Fayoumi, and Saeed-Ul Hassan. "Bot prediction on social networks of Twitter in altmetrics using deep graph convolutional networks." Soft Computing (2020): 1-12.
-Hopwood, Michael, Phuong Pho, and Alexander V. Mantzaris. "Exploring the Value of Nodes with Multicommunity Membership for Classification with Graph Convolutional Neural Networks." Information 12.4 (2021): 170.
l.58, SGC models can treat the information in a non equal setting
-Wu, Felix, et al. "Simplifying graph convolutional networks." International conference on machine learning. PMLR, 2019. -Utilizing the simple graph convolutional neural network as a model for simulating influence spread in networks AV Mantzaris, D Chiodini, K Ricketson Computational Social Networks 8 (1), 1-17 l.97 utilizeseqn 3, what is W^k_{pool} ? since it is a key premise of your methodology please place it as an equation.
For Figure 8 can you apply a spectral clustering approach (based upon the Laplacian) and DB scan or a related measure on those graphs from T-SNE? Then compare the coloring with those you produced.
At the start of Section 4, can you describe again, for the readers' convenience' how you measure the discrepancy between the true popularity values of a cascade found in the data and that you predict? (what is the 'loss' function your methodology works to minimize)
Author Response
Thanks for your comments, I have made the correction. Please see the attachment.

Reviewer 2 Report
The article deals with the problem of predicting the popularity of online content, especially that in social networks by the employment of machine learning techniques and on particular deep learning networks. In particular, the authors propose a novel framework integrating the cascade graph information and temporal information to predict the future popularity of online contents. Of interest is the novel graph pooling strategy employed, while the inter-infection duration time information is incorporated into the model by using Long Short Term Memory network(LSTM). The authors present a set of carefully perfοrmed experiments , on two real datasets that lead to the interesting conclusion that temporal information is a better predictor for popularity prediction than cascade graph information. Perhaps this is the most interesting
The article is very well written and the topic dealed with is one of the most interesting in the area of computer science these days. Despite the existence of a lot of works with similar content it contains ideas of interest to prospective researchers in the area. The main techniques seem to be novel and modern while the experiments well worked out. On the other hand the authors should explain in more detail the technical points in the described methodology where the article deviates from similar approaches either in the algorithmic concepts or in the practical techniques used when developing the system. Moreover, it is imperative to have more details concerning extensions and future work in the proposed system.
Author Response
Thank you for your advice. I have added some content in l.199 and in l.269 to emphasis on the differences between our proposed method and other similar methods. I add some content to compare our work with other similar approach in l.114. I write more details concerning extensions and future work in l.476 in section 5 “Conclusion and Future Work”.
Reviewer 3 Report
In this manuscript, the authors propose a novel deep learning model for the popularity prediction of online contents, that exploits an efficient graph pooling method for generating cascade graph representation. To assess the effectiveness of their model, the authors also provide a strong experimental evaluation on two real-world datasets, and a comparison with some state-of-the-art methods.
The work is very well written, well organised and of definite interest to the scientific community. Both the background, the definition of the problem and description of the approach, and (above all) the experimental validation, are extremely rigorous in terms of scientific goodness.
I believe that this work is therefore worthy of publication in this prestigious journal. I have only one minor observation, concerning the fact that the authors - in their description of the related work - have focused on Facebook, Twitter, Sina Weibo and TikTok, neglecting to mention (at least) some important recent work in the field of popularity prediction on other leading social networks, for example (not exhaustively):
- Instagram: https://doi.org/10.3390/info11090453;
- Youtube: https://doi.org/10.1145/3132847.3132997;
- Twitch: https://doi.org/10.1109/BigComp48618.2020.00-84
I strongly suggest, therefore, that the authors consolidate the references so as to generalise the context in which this very good work is situated.
Author Response
Thank you for your advice. I have added more content in l.102 section 2 “Background and Related Works” to reference some important recent works in the field of popularity prediction.
Reviewer 4 Report
The proposed work, titled “Popularity Prediction of Online Contents via Cascade Graph and Temporal Information” has been focused on the prediction of the popularity of online content.
In such a context, the authors propose an approach aimed to face the well-known problems of the state-of-the-art solutions, which is based on a LSTM neural network.
On the basis of the performed experiments on real-world datasets, they claim that the proposed approach overcomes the state-of-the-art solutions in terms of prediction accuracy.
I suggest to the authors a careful re-reading of the entire manuscript in order to fix some minor typos (e.g., "two reshare event", “representaion”) and grammar forms (e.g., correct use of while/whereas/meanwhile,, that/which, and so on).
The "Introduction" section of the manuscript appears unbalanced compared to the "Related Work": I suggest using the “Introduction” section only for a brief overview of the context taken into account, expanding each concept by using the "Related Work" section.
In addition, the “Introduction” and (above all) the “Related Work” (that I suggest renaming as “Background and Related Works”) sections should cite and discuss further papers focused on information very close or directly related to the research field taken into account, i.e., the authors should add more in-deep information and additional literature references.
The references do not appear very updated, so the authors should check if there are more recent works among those he mentioned and, in any case, according to my previous consideration the “Background and Related Works” should be extended citing and discussing additional literature works close or directly related to the considered domain, e.g.:
(-) Adewole, K. S., Han, T., Wu, W., Song, H., \& Sangaiah, A. K. (2020). Twitter spam account detection based on clustering and classification methods. The Journal of Supercomputing, 76(7), 4802-4837.
(-) Roberto, Saia, et al. A Supervised Multi-class Multi-label Word Embeddings Approach for Toxic Comment Classification. In: KDIR. 2019. p. 105-112.
(-) Ma, Tinghuai, et al. "A novel rumor detection algorithm based on entity recognition, sentence reconfiguration, and ordinary differential equation network." Neurocomputing 447 (2021): 224-234.
(-) Ma, Tinghuai, et al. "A Comprehensive Trust Model Based on Social Relationship and Transaction Attributes." Security and Communication Networks 2020 (2020).
(-) Carta, Salvatore, et al. "Popularity prediction of instagram posts." Information 11.9 (2020): 453.
(-) … And so on.
The formal approach followed by the author and the related experimental results have been presented in a quite clear form to the readers.
About the “Conclusions” section (that should be renamed to “Conclusions and Future Work”, considering that it also includes the future research directions of the authors), the authors should expand the given information by recapping in brief all the main steps of their manuscript, in order to offer to the readers a brief but complete summary of the work carried out.
In this section, the authors should also better underline the possible advantages of the proposed approach with regard to both the state-of-the-art solutions and the real-world scenarios.
Summarizing, apart from the minor issues I highlighted above, the main weakness I recognized in the proposed work is the absence of a “clear” and “strong” scientific contribution: for this reason, the authors should better highlight their scientific contribution, with regard to the current literature (e.g., Cao, Qi, et al. "Popularity prediction on social platforms with coupled graph neural networks." Proceedings of the 13th International Conference on Web Search and Data Mining. 2020., and similar ones.), also in light of real use cases.
Author Response

(The authors gave the same response as above.)
